# Residual Force Control for Agile Human Behavior Imitation and Extended Motion Synthesis

**Ye Yuan**      **Kris M. Kitani**
Robotics Institute
Carnegie Mellon University
{yyuan2, kkitani}@cs.cmu.edu

## Abstract

Reinforcement learning has shown great promise for synthesizing realistic human behaviors by learning humanoid control policies from motion capture data. However, it is still very challenging to reproduce sophisticated human skills like ballet dance, or to stably imitate long-term human behaviors with complex transitions. The main difficulty lies in the *dynamics mismatch* between the humanoid model and real humans. That is, motions of real humans may not be physically possible for the humanoid model. To overcome the dynamics mismatch, we propose a novel approach, *residual force control (RFC)*, that augments a humanoid control policy by adding external residual forces into the action space. During training, the RFC-based policy learns to apply residual forces to the humanoid to compensate for the dynamics mismatch and better imitate the reference motion. Experiments on a wide range of dynamic motions demonstrate that our approach outperforms state-of-the-art methods in terms of convergence speed and the quality of learned motions. Notably, we showcase a physics-based virtual character empowered by RFC that can perform highly agile ballet dance moves such as pirouette, arabesque and jeté. Furthermore, we propose a dual-policy control framework, where a kinematic policy and an RFC-based policy work in tandem to synthesize multimodal infinite-horizon human motions without any task guidance or user input. Our approach is the first humanoid control method that successfully learns from a large-scale human motion dataset (Human3.6M) and generates diverse long-term motions. Code and videos are available at https://www.ye-yuan.com/rfc.

## 1 Introduction

Understanding human behaviors and creating virtual humans that act like real people has been a mesmerizing yet elusive goal in computer vision and graphics. One important step to achieve this goal is human motion synthesis which has broad applications in animation, gaming and virtual reality. With advances in deep learning, data-driven approaches [12, 13, 41, 38, 3] have achieved remarkable progress in producing realistic motions learned from motion capture data. Among them are physics-based methods [41, 38, 3] empowered by reinforcement learning (RL), where a humanoid agent in simulation is trained to imitate reference motions. Physics-based methods have many advantages over their kinematics-based counterparts. For instance, the motions generated with physics are typically free from jitter, foot skidding or geometry penetration as they respect physical constraints. Moreover, the humanoid agent inside simulation can interact with the physical environment and adapt to various terrains and perturbations, generating diverse motion variations.

However, physics-based methods have their own challenges. In many cases, the humanoid agent fails to imitate highly agile motions like ballet dance or long-term motions that involve swift transitions between various locomotions. We attribute such difficulty to the *dynamics mismatch* between the humanoid model and real humans. Humans are very difficult to model because they are very complex

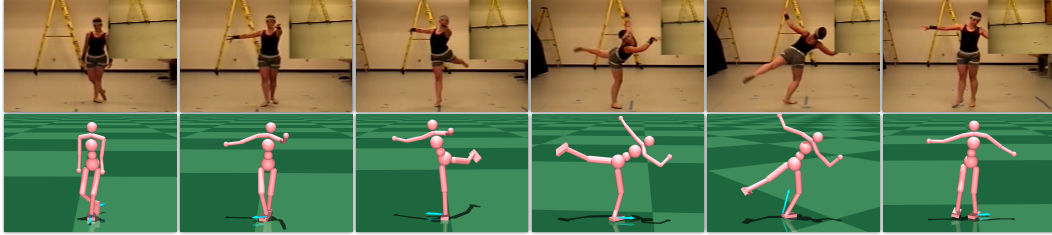

Figure 1: **Top:** A ballet dancer performing highly agile moves like jeté, arabesque and pirouette. **Bottom:** A humanoid agent controlled by a policy augmented with the proposed residual forces (blue arrows) is able to dance like the performer. The motion is best viewed in our supplementary video.

creatures with hundreds of bones and muscles. Although prior work has tried to improve the fidelity of the humanoid model [21, 52], it is nonetheless safe to say that these models are not exact replicas of real humans and the dynamics mismatch still exists. The problem is further complicated when motion capture data comprises a variety of individuals with diverse body types. Due to the dynamics mismatch, motions produced by real humans may not be admissible by the humanoid model, which means no control policy of the humanoid is able to generate those motions.

To overcome the dynamics mismatch, we propose an approach termed *Residual Force Control (RFC)* which can be seamlessly integrated into existing RL-based humanoid control frameworks. Specifically, RFC augments a control policy by introducing external residual forces into the action space. During RL training, the RFC-based policy learns to apply residual forces onto the humanoid to compensate for the dynamics mismatch and achieve better motion imitation. Intuitively, the residual forces can be interpreted as invisible forces that enhance the humanoid's abilities to go beyond the physical limits imposed by the humanoid model. RFC generates a more expressive dynamics that admits a wider range of human motions since the residual forces serve as a learnable time-varying correction to the dynamics of the humanoid model. To validate our approach, we perform motion imitation experiments on a wide range of dynamic human motions including ballet dance and acrobatics. The results demonstrate that RFC outperforms state-of-the-art methods with faster convergence and better motion quality. Notably, we are able to showcase humanoid control policies that are capable of highly agile ballet dance moves like pirouette, arabesque and jeté (Fig. 1).

Another challenge facing physics-based methods is synthesizing multi-modal long-term human motions. Previous work has elicited long-term human motions with hierarchical RL [33, 32, 42] or user interactive control [3, 38]. However, these approaches still need to define high-level tasks of the agent or require human interaction. We argue that removing these requirements could be critical for applications like automated motion generation and large-scale character animation. Thus, we take a different approach to long-term human motion synthesis by leveraging the temporal dependence of human motion. In particular, we propose a *dual-policy control* framework where a kinematic policy learns to predict multi-modal future motions based on the past motion and a latent variable used to model human intent, while an RFC-based control policy learns to imitate the output motions of the kinematic policy to produce physically-plausible motions. Experiments on a large-scale human motion dataset, Human3.6M [14], show that our approach with RFC and dual policy control can synthesize stable long-term human motions without any task guidance or user input.

The main contributions of this work are as follows: (1) We address the dynamics mismatch in motion imitation by introducing the idea of RFC which can be readily integrated into RL-based humanoid control frameworks. (2) We propose a dual-policy control framework to synthesize multi-modal long-term human motions without the need for task guidance or user input. (3) Extensive experiments show that our approach outperforms state-of-the-art methods in terms of learning speed and motion quality. It also enables imitating highly agile motions like ballet dance that evade prior work. With RFC and dual-policy control, we present the first humanoid control method that successfully learns from a large-scale human motion dataset (Human3.6M) and generates diverse long-term motions.

## 2 Related Work

**Kinematics-based models** for human motion synthesis have been extensively studied by the computer graphics community. Early approaches construct motion graphs from large motion datasets and

design controllers to navigate through the graph to generate novel motions [20, 43]. Alternatively, prior work has explored learning a low-dimensional embedding space to synthesize motions continuously [45, 23, 24]. Advances in deep learning have enabled methods that use deep neural networks to design generative models of human motions [12, 13, 47]. While the graphics community focuses on user control, computer vision researchers have been increasingly interested in predicting future human motions. A vast body of work has used recurrent neural networks to predict a deterministic future motion from the past motion [9, 16, 25, 30, 50, 39, 1, 10]. To address the uncertainty of future, stochastic approaches develop deep generative models to predict multi-modal future motions [53, 2, 19, 56, 58]. The major drawback of kinematics-based approaches is that they are prone to generating physically-invalid motions with artifacts like jitter, foot skidding and geometry (e.g., body, ground) penetration.

**Physics-based methods** for motion synthesis address the limitation of kinematics-based models by enforcing physical constraints. Early work has adopted model-based methods for tracking reference motions [54, 36, 22, 28, 29]. Recently, deep RL has achieved great success in imitating human motions with manually-designed rewards [26, 27, 41]. GAIL [11] based approaches have been proposed to eliminate the need for reward engineering [31, 51]. RL-based humanoid control has also been applied to estimating physically-plausible human poses from videos [55, 57, 15]. To synthesize long-term human motions, prior work has resorted to hierarchical RL with predefined high-level task objectives [33, 32, 42]. Alternatively, recent works use deep RL to learn controllable polices to generate long-term motions with user input [3, 38]. Different from previous work, our dual-policy control framework exploits the temporal dependence of human motion and synthesizes multi-modal long-term motions by forecasting diverse futures, which can be used to replace manual task guidance or user input. Furthermore, our proposed residual force control addresses the dynamics mismatch in humanoid control and enables imitating agile motions like ballet dance that evade prior work.

**Inferring external forces** from human motion has been an active research area in biomechanics. Researchers have developed models that regress ground reaction forces from human motion using supervised learning [4, 37, 17]. These approaches require expensive force data collected in laboratory settings to train the models. On the other hand, machine learning researchers have proposed differentiable physics engines that enable learning forces to control simple simulated systems [6, 7]. Trajectory optimization based approaches [35, 34] have also been used to optimize external contact forces to synthesize human motions. A recent work [8] predicts forces acting on rigid objects in simulation to match image evidence with contact point supervision. Unlike prior work, we use deep RL to learn residual forces that complement contact forces to improve motion imitation without the supervision of forces or contact points.

## 3 Preliminaries

The task of humanoid control-based motion imitation can be formulated as a Markov decision process (MDP), which is defined by a tuple $\mathcal{M} = (\mathcal{S}, \mathcal{A}, \mathcal{T}, R, \gamma)$ of states, actions, transition dynamics, a reward function, and a discount factor. A humanoid agent interacts with a physically-simulated environment according to a policy $\pi(\boldsymbol{a}|\boldsymbol{s})$, which models the conditional distribution of choosing an action $\boldsymbol{a} \in \mathcal{A}$ given the current state $\boldsymbol{s} \in \mathcal{S}$. Starting from some initial state $\boldsymbol{s}_0$, the agent iteratively samples an action $\boldsymbol{a}_t$ from the policy $\pi$ and the simulation environment with transition dynamics $\mathcal{T}(\boldsymbol{s}_{t+1}|\boldsymbol{s}_t, \boldsymbol{a}_t)$ generates the next state $\boldsymbol{s}_{t+1}$ and gives the agent a reward $r_t$. The reward is assigned based on how the agent's motion aligns with a given reference motion. The agent's goal is to learn an optimal policy $\pi^*$ that maximizes its expected return $J(\pi) = \mathbb{E}_{\pi}\left[\sum_t \gamma^t r_t\right]$. To solve for the optimal policy, one can apply one's favorite reinforcement learning algorithm (e.g., PPO [44]). In the following, we will give a more detailed description of the states, actions, policy and rewards to show how motion imitation fits in the standard reinforcement learning (RL) framework.

**States.** The state $\boldsymbol{s}$ is formed by the humanoid state $\boldsymbol{x} = (\boldsymbol{q}, \dot{\boldsymbol{q}})$ which includes all degrees of freedom (DoFs) $\boldsymbol{q}$ of the humanoid and their corresponding velocities $\dot{\boldsymbol{q}}$. Specifically, the DoFs $\boldsymbol{q} = (\boldsymbol{q}_{\mathrm{r}}, \boldsymbol{q}_{\mathrm{nr}})$ include 6 root DoFs $\boldsymbol{q}_{\mathrm{r}}$ (global position and orientation) as well as the angles of other joints $\boldsymbol{q}_{\mathrm{nr}}$. We transform $\boldsymbol{q}_{\mathrm{r}}$ to the root's local coordinate to remove dependency on global states.

**Actions.** As noticed in previous work [40, 57], using proportional derivative (PD) controllers at each joint yields more robust policies than directly outputting joint torques. Thus, the action $\boldsymbol{a}$ consists of the target angles $\boldsymbol{u}$ of the PD controllers mounted at non-root joint DoFs $\boldsymbol{q}_{\mathrm{nr}}$ (root DoFs $\boldsymbol{q}_{\mathrm{r}}$ are not

actuated). The joint torques $\boldsymbol{\tau}$ can then be computed as

$$\boldsymbol{\tau} = \boldsymbol{k}_{\mathrm{p}} \circ (\boldsymbol{u} - \boldsymbol{q}_{\mathrm{nr}}) - \boldsymbol{k}_{\mathrm{d}} \circ \dot{\boldsymbol{q}}_{\mathrm{nr}}, \tag{1}$$

where $\boldsymbol{k}_{\mathrm{p}}$ and $\boldsymbol{k}_{\mathrm{d}}$ are manually-specified gains and $\circ$ denotes element-wise multiplication.

**Policy.** As the action $\boldsymbol{a}$ is continuous, we use a parametrized Gaussian policy $\pi_\theta(\boldsymbol{a}|\boldsymbol{s}) = \mathcal{N}(\boldsymbol{\mu}_\theta, \boldsymbol{\Sigma})$ where the mean $\boldsymbol{\mu}_\theta$ is output by a neural network with parameters $\theta$ and $\boldsymbol{\Sigma}$ is a fixed diagonal covariance matrix. At test time, instead of sampling we use the mean action to achieve best performance.

**Rewards.** Given a reference motion $\widehat{\boldsymbol{x}}_{0:T} = (\widehat{\boldsymbol{x}}_0, \ldots, \widehat{\boldsymbol{x}}_{T-1})$, we need to design a reward function to incentivize the humanoid agent to imitate $\widehat{\boldsymbol{x}}_{0:T}$. To this end, the reward $r_t = r_t^{\mathrm{im}}$ is defined by an imitation reward $r_t^{\mathrm{im}}$ that encourages the state $\boldsymbol{x}_t$ of the humanoid agent to match the reference state $\widehat{\boldsymbol{x}}_t$. The detailed definition of the imitation reward $r_t^{\mathrm{im}}$ can be found in the supplementary materials.

During RL training, the agent's initial state is intialized to a random frame from the reference motion $\widehat{\boldsymbol{x}}_{0:T}$. The episode ends when the agent falls to the ground or the episode horizon $H$ is reached.

## 4  Residual Force Control (RFC)

As demonstrated in prior work [41, 57], we can apply the motion imitation framework described in Sec. 3 to successfully learn control policies that imitate human locomotions (e.g., walking, running, crouching) or acrobatics (e.g, backflips, cartwheels, jump kicks). However, the motion imitation framework has its limit on the range of motions that the agent is able to imitate. In our experiments, we often find the framework unable to learn more complex motions that require sophisticated foot interaction with the ground (e.g., ballet dance) or long-term motions that involve swift transitions between different modes of locomotion. We posit that the difficulty in learning such highly agile motions can be attributed to the *dynamics mismatch* between the humanoid model and real humans, i.e., the humanoid transition dynamics $\mathcal{T}(\boldsymbol{s}_{t+1}|\boldsymbol{s}_t, \boldsymbol{a}_t)$ is different from the real human dynamics. Thus, due to the dynamics mismatch, a reference motion $\widehat{\boldsymbol{x}}_{0:T}$ generated by a real human may not be admissible by the transition dynamics $\mathcal{T}$, which means no policy under $\mathcal{T}$ can generate $\widehat{\boldsymbol{x}}_{0:T}$.

To overcome the dynamics mismatch, our goal is to come up with a new transition dynamics $\mathcal{T}'$ that admits a wider range of motions. The new transition dynamics $\mathcal{T}'$ should ideally satisfy two properties: (1) $\mathcal{T}'$ needs to be expressive and overcome the limitations of the current dynamics $\mathcal{T}$; (2) $\mathcal{T}'$ needs to be physically-valid and respect physical constraints (e.g., contacts), which implies that kinematics-based approaches such as directly manipulating the resulting state $\boldsymbol{s}_{t+1}$ by adding some residual $\delta\boldsymbol{s}$ are not viable as they may violate physical constraints.

Based on the above considerations, we propose *residual force control (RFC)*, that considers a more general form of dynamics $\widetilde{\mathcal{T}}(\boldsymbol{s}_{t+1}|\boldsymbol{s}_t, \boldsymbol{a}_t, \widetilde{\boldsymbol{a}}_t)$ where we introduce a corrective control action $\widetilde{\boldsymbol{a}}_t$ (i.e., external residual forces acting on the humanoid) alongside the original humanoid control action $\boldsymbol{a}_t$. We also introduce a corresponding RFC-based composite policy $\widetilde{\pi}_\theta(\boldsymbol{a}_t, \widetilde{\boldsymbol{a}}_t|\boldsymbol{s}_t)$ which can be decomposed into two policies: (1) the original policy $\widetilde{\pi}_{\theta_1}(\boldsymbol{a}_t|\boldsymbol{s}_t)$ with parameters $\theta_1$ for humanoid control and (2) a residual force policy $\widetilde{\pi}_{\theta_2}(\widetilde{\boldsymbol{a}}_t|\boldsymbol{s}_t)$ with parameters $\theta_2$ for corrective control. The RFC-based dynamics and policy are more general as the original policy $\widetilde{\pi}_{\theta_1}(\boldsymbol{a}_t|\boldsymbol{s}_t) \equiv \widetilde{\pi}_\theta(\boldsymbol{a}_t, \boldsymbol{0}|\boldsymbol{s}_t)$ corresponds to a policy $\widetilde{\pi}_\theta$ that always outputs zero residual forces. Similarly, the original dynamics $\mathcal{T}(\boldsymbol{s}_{t+1}|\boldsymbol{s}_t, \boldsymbol{a}_t) \equiv \widetilde{\mathcal{T}}(\boldsymbol{s}_{t+1}|\boldsymbol{s}_t, \boldsymbol{a}_t, \boldsymbol{0})$ corresponds to the dynamics $\widetilde{\mathcal{T}}$ with zero residual forces. During RL training, the RFC-based policy $\widetilde{\pi}_\theta(\boldsymbol{a}_t, \widetilde{\boldsymbol{a}}_t|\boldsymbol{s}_t)$ learns to apply proper residual forces $\widetilde{\boldsymbol{a}}_t$ to the humanoid to compensate for the dynamics mismatch and better imitate the reference motion. Since $\widetilde{\boldsymbol{a}}_t$ is sampled from $\widetilde{\pi}_{\theta_2}(\widetilde{\boldsymbol{a}}_t|\boldsymbol{s}_t)$, the dynamics of the original policy $\widetilde{\pi}_{\theta_1}(\boldsymbol{a}_t|\boldsymbol{s}_t)$ is parametrized by $\theta_2$ as $\mathcal{T}'_{\theta_2}(\boldsymbol{s}_{t+1}|\boldsymbol{s}_t, \boldsymbol{a}_t) \equiv \widetilde{\mathcal{T}}(\boldsymbol{s}_{t+1}|\boldsymbol{s}_t, \boldsymbol{a}_t, \widetilde{\boldsymbol{a}}_t)$. From this perspective, $\widetilde{\boldsymbol{a}}_t$ are learnable time-varying dynamics correction forces governed by $\widetilde{\pi}_{\theta_2}$. Thus, by optimizing the composite policy $\widetilde{\pi}_\theta(\boldsymbol{a}_t, \widetilde{\boldsymbol{a}}_t|\boldsymbol{s}_t)$, we are in fact jointly optimizing the original humanoid control action $\boldsymbol{a}_t$ and the dynamics correction (residual forces) $\widetilde{\boldsymbol{a}}_t$. In the following, we propose two types of RFC, each with its own advantages.

### 4.1  RFC-Explicit

One way to implement RFC is to explicitly model the corrective action $\widetilde{\boldsymbol{a}}_t$ as a set of residual force vectors $\{\boldsymbol{\xi}_1, \ldots, \boldsymbol{\xi}_M\}$ and their respective contact points $\{\boldsymbol{e}_1, \ldots, \boldsymbol{e}_M\}$. As the humanoid model is formed by a set of rigid bodies, the residual forces are applied to $M$ bodies of the humanoid, where

$\boldsymbol{\xi}_j$ and $\boldsymbol{e}_j$ are represented in the local body frame. To reduce the size of the corrective action space, one can apply residual forces to a limited number of bodies such as the hip or feet. In RFC-Explicit, the corrective action of the policy $\widetilde{\pi}_\theta(\boldsymbol{a}, \widetilde{\boldsymbol{a}}|\boldsymbol{s})$ is defined as $\widetilde{\boldsymbol{a}} = (\boldsymbol{\xi}_1, \ldots, \boldsymbol{\xi}_M, \boldsymbol{e}_1, \ldots, \boldsymbol{e}_M)$ and the humanoid control action is $\boldsymbol{a} = \boldsymbol{u}$ as before (Sec. 3). We can describe the humanoid motion using the equation of motion for multibody systems [46] augmented with the proposed residual forces:

$$\boldsymbol{B}(\boldsymbol{q})\ddot{\boldsymbol{q}} + \boldsymbol{C}(\boldsymbol{q}, \dot{\boldsymbol{q}})\dot{\boldsymbol{q}} + \boldsymbol{g}(\boldsymbol{q}) = \begin{bmatrix} \boldsymbol{0} \\ \boldsymbol{\tau} \end{bmatrix} + \underbrace{\sum_i \boldsymbol{J}_{\boldsymbol{v}_i}^T \boldsymbol{h}_i}_{\text{Contact Forces}} + \underbrace{\sum_{j=1}^M \boldsymbol{J}_{\boldsymbol{e}_j}^T \boldsymbol{\xi}_j}_{\textbf{Residual Forces}} , \qquad (2)$$

where we have made the residual forces term explicit. Eq. (2) is an ordinary differential equation (ODE), and by solving it with an ODE solver we obtain the aforementioned RFC-based dynamics $\widetilde{\mathcal{T}}(\boldsymbol{s}_{t+1}|\boldsymbol{s}_t, \boldsymbol{a}_t, \widetilde{\boldsymbol{a}}_t)$. On the left hand side $\ddot{\boldsymbol{q}}, \boldsymbol{B}, \boldsymbol{C}, \boldsymbol{g}$ are the joint accelerations, the inertial matrix, the matrix of Coriolis and centrifugal terms, and the gravity vector, respectively. On the right hand side, the first term contains the torques $\boldsymbol{\tau}$ computed from $\boldsymbol{a}$ (Sec. 3) applied to the non-root joint DoFs $\boldsymbol{q}_{\text{nr}}$ and $\boldsymbol{0}$ corresponds to the 6 non-actuated root DoFs $\boldsymbol{q}_{\mathbf{r}}$. The second term involves existing contact forces $\boldsymbol{h}_i$ on the humanoid (usually exerted by the ground plane) and the contact points $\boldsymbol{v}_i$ of $\boldsymbol{h}_i$, which are determined by the simulation environment. Here, $\boldsymbol{J}_{\boldsymbol{v}_i} = d\boldsymbol{v}_i/d\boldsymbol{q}$ is the Jacobian matrix that describes how the contact point $\boldsymbol{v}_i$ changes with the joint DoFs $\boldsymbol{q}$. By multiplying $\boldsymbol{J}_{\boldsymbol{v}_i}^T$, the contact force $\boldsymbol{h}_i$ is transformed from the world space to the joint space, which can be understood using the principle of virtual work, i.e., the virtual work in the joint space equals that in the world space or $(\boldsymbol{J}_{\boldsymbol{v}_i}^T \boldsymbol{h}_i)^T d\boldsymbol{q} = \boldsymbol{h}_i^T d\boldsymbol{v}_i$. Unlike the contact forces $\boldsymbol{h}_i$ which are determined by the environment, the policy can control the corrective action $\widetilde{\boldsymbol{a}}$ which includes the residual forces $\boldsymbol{\xi}_j$ and their contact points $\boldsymbol{e}_j$ in the proposed third term. The Jacobian matrix $\boldsymbol{J}_{\boldsymbol{e}_j} = d\boldsymbol{e}_j/d\boldsymbol{q}$ is similarly defined as $\boldsymbol{J}_{\boldsymbol{v}_i}$. During RL training, the policy will learn to adjust $\boldsymbol{\xi}_j$ and $\boldsymbol{e}_j$ to better imitate the reference motion. Most popular physics engines (e.g., MuJoCo [49], Bullet [5]) use a similar equation of motion to Eq. (2) (without residual forces), which makes our approach easy to integrate.

As the residual forces are designed to be a correction mechanism to the original humanoid dynamics $\mathcal{T}$, we need to regularize the residual forces so that the policy only invokes the residual forces when necessary. Consequently, the regularization keeps the new dynamics $\mathcal{T}'$ close to the original dynamics $\mathcal{T}$. Formally, we change the RL reward function by adding a regularizing reward $r_t^{\texttt{reg}}$:

$$r_t = r_t^{\texttt{im}} + w_{\texttt{reg}} r_t^{\texttt{reg}}, \quad r_t^{\texttt{reg}} = \exp\left(-\sum_{j=1}^M \left(k_{\texttt{f}} \left\|\boldsymbol{\xi}_j\right\|^2 + k_{\texttt{cp}} \left\|\boldsymbol{e}_j\right\|^2\right)\right), \qquad (3)$$

where $w_{\texttt{reg}}, k_{\texttt{f}}$ and $k_{\texttt{cp}}$ are weighting factors. The regularization constrains the residual force $\boldsymbol{\xi}_j$ to be as small as possible and pushes the contact point $\boldsymbol{e}_j$ to be close to the local body origin.

## 4.2 RFC-Implicit

One drawback of RFC-explicit is that one must specify the number of residual forces and the contact points. To address this issue, we also propose an implicit version of RFC where we directly model the total joint torques $\boldsymbol{\eta} = \sum \boldsymbol{J}_{\boldsymbol{e}_j}^T \boldsymbol{\xi}_j$ of the residual forces. In this way, we do not need to specify the number of residual forces or the contact points. We can decompose $\boldsymbol{\eta}$ into two parts $(\boldsymbol{\eta}_{\mathbf{r}}, \boldsymbol{\eta}_{\text{nr}})$ that correspond to the root and non-root DoFs respectively. We can merge $\boldsymbol{\eta}$ with the first term on the right of Eq. (2) as they are both controlled by the policy, which yields the new equation of motion:

$$\boldsymbol{B}(\boldsymbol{q})\ddot{\boldsymbol{q}} + \boldsymbol{C}(\boldsymbol{q}, \dot{\boldsymbol{q}})\dot{\boldsymbol{q}} + \boldsymbol{g}(\boldsymbol{q}) = \begin{bmatrix} \boldsymbol{\eta}_{\mathbf{r}} \\ \boldsymbol{\tau} \pm \boldsymbol{\eta}_{\text{nr}} \end{bmatrix} + \sum_i \boldsymbol{J}_{\boldsymbol{v}_i}^T \boldsymbol{h}_i, \qquad (4)$$

where we further remove $\boldsymbol{\eta}_{\text{nr}}$ (crossed out) because the torques applied at non-root DoFs are already modeled by the policy $\widetilde{\pi}_\theta(\boldsymbol{a}, \widetilde{\boldsymbol{a}}|\boldsymbol{s})$ through $\boldsymbol{\tau}$ which can absorb $\boldsymbol{\eta}_{\text{nr}}$. In RFC-Implicit, the corrective action of the policy is defined as $\widetilde{\boldsymbol{a}} = \boldsymbol{\eta}_{\mathbf{r}}$. To regularize $\boldsymbol{\eta}_{\mathbf{r}}$, we use a similar reward to Eq. (3):

$$r_t = r_t^{\texttt{im}} + w_{\texttt{reg}} r_t^{\texttt{reg}}, \quad r_t^{\texttt{reg}} = \exp\left(-k_{\mathbf{r}} \left\|\boldsymbol{\eta}_{\mathbf{r}}\right\|^2\right), \qquad (5)$$

where $k_{\mathbf{r}}$ is a weighting factor. While RFC-Explicit provides more interpretable results by exposing the residual forces and their contact points, RFC-Implicit is computationally more efficient as it only

increases the action dimensions by 6 which is far less than that of RFC-Explicit and it does not require Jacobian computation. Furthermore, RFC-Implicit does not make any underlying assumptions about the number of residual forces or their contact points.

## 5 Dual-Policy Control for Extended Motion Synthesis

So far our focus has been on imitating a given reference motion, which in practice is typically a short and segmented motion capture sequence (e.g., within 10 seconds). In some applications (e.g., behavior simulation, large-scale animation), we want the humanoid agent to autonomously exhibit long-term behaviors that consist of a sequence of diverse agile motions. Instead of guiding the humanoid using manually-designed tasks or direct user input, our goal is to let the humanoid learn long-term behaviors directly from data. To achieve this, we need to develop an approach that (i) infers future motions from the past and (ii) captures the multi-modal distribution of the future.

As multi-modal behaviors are usually difficult to model in the control space due to non-differentiable dynamics, we first model human behaviors in the kinematic space. We propose a *dual-policy control* framework that consists of a kinematic policy $\kappa_\psi$ and an RFC-based control policy $\widetilde{\pi}_\theta$. The $\psi$-parametrized kinematic policy $\kappa_\psi(\boldsymbol{x}_{t:t+f}|\boldsymbol{x}_{t-p:t}, \boldsymbol{z})$ models the conditional distribution over a $f$-step future motion $\boldsymbol{x}_{t:t+f}$, given a $p$-step past motion $\boldsymbol{x}_{t-p:t}$ and a latent variable $\boldsymbol{z}$ used to model human intent. We learn the kinematic policy $\kappa_\psi$ with a conditional variational autoencoder (CVAE [18]), where we optimize the evidence lower bound (ELBO):

$$\mathcal{L} = \mathbb{E}_{q_\phi(\boldsymbol{z}|\boldsymbol{x}_{t-p:t}, \boldsymbol{x}_{t:t+f})}\left[\log \kappa_\psi(\boldsymbol{x}_{t:t+f}|\boldsymbol{x}_{t-p:t}, \boldsymbol{z})\right] - \text{KL}\left(q_\phi(\boldsymbol{z}|\boldsymbol{x}_{t-p:t}, \boldsymbol{x}_{t:t+f})\|p(\boldsymbol{z})\right), \quad (6)$$

where $q_\phi(\boldsymbol{z}|\boldsymbol{x}_{t-p:t}, \boldsymbol{x}_{t:t+f})$ is a $\phi$-parametrized approximate posterior (encoder) distribution and $p(\boldsymbol{z})$ is a Gaussian prior. The kinematic policy $\kappa_\psi$ and encoder $q_\phi$ are instantiated as Gaussian distributions whose parameters are generated by two recurrent neural networks (RNNs) respectively. The detailed architectures for $\kappa_\psi$ and $q_\phi$ are given in the supplementary materials.

Once the kinematic policy $\kappa_\psi$ is learned, we can generate multi-modal future motions $\widehat{\boldsymbol{x}}_{t:t+f}$ from the past motion $\boldsymbol{x}_{t-p:t}$ by sampling $\boldsymbol{z} \sim p(\boldsymbol{z})$ and decoding $\boldsymbol{z}$ with $\kappa_\psi$. To produce physically-plausible motions, we use an RFC-based control policy $\widetilde{\pi}_\theta(\boldsymbol{a}, \widetilde{\boldsymbol{a}}|\boldsymbol{x}, \widehat{\boldsymbol{x}}, \boldsymbol{z})$ to imitate the output motion $\widehat{\boldsymbol{x}}_{t:t+f}$ of $\kappa_\psi$ by treating $\widehat{\boldsymbol{x}}_{t:t+f}$ as the reference motion in the motion imitation framework (Sec. 3 and 4). The state $\boldsymbol{s}$ of the policy now includes the state $\boldsymbol{x}$ of the humanoid, the reference state $\widehat{\boldsymbol{x}}$ from $\kappa_\psi$, and the latent code $\boldsymbol{z}$. To fully leverage the reference state $\widehat{\boldsymbol{x}}$, we use the non-root joint angles $\widehat{\boldsymbol{q}}_{\text{nr}}$ inside $\widehat{\boldsymbol{x}}$ to serve as bases for the target joint angles $\boldsymbol{u}$ of the PD controllers. For this purpose, we change the humanoid control action $\boldsymbol{a}_t$ from $\boldsymbol{u}$ to residual angles $\delta\boldsymbol{u}$, and $\boldsymbol{u}$ can be computed as $\boldsymbol{u} = \widehat{\boldsymbol{q}}_{\text{nr}} + \delta\boldsymbol{u}$. This additive action will improve policy learning because $\widehat{\boldsymbol{q}}_{\text{nr}}$ provides a good guess for $\boldsymbol{u}$.

---

**Algorithm 1** Learning RFC-based policy $\widetilde{\pi}_\theta$ in dual-policy control

---

1: **Input:** motion data $\mathcal{X}$, pretrained kinematic policy $\kappa_\psi$
2: $\theta \leftarrow$ random weights
3: **while** not converged **do**
4:      $\mathcal{D} \leftarrow \emptyset$                                                    ▷ initialize sample memory
5:      **while** $\mathcal{D}$ is not full **do**
6:          $\widehat{\boldsymbol{x}}_{0:p} \leftarrow$ random motion from $\mathcal{X}$
7:          $\boldsymbol{x}_{p-1} \leftarrow \widehat{\boldsymbol{x}}_{p-1}$                                    ▷ initialize humanoid state
8:          **for** $t \leftarrow p, \ldots, p + nf - 1$ **do**
9:              **if** $(t - p) \bmod f = 0$ **then**          ▷ if reaching end of reference motion segment
10:                 $\boldsymbol{z} \sim p(\boldsymbol{z})$
11:                 $\widehat{\boldsymbol{x}}_{t:t+f} \leftarrow \kappa_\psi(\widehat{\boldsymbol{x}}_{t:t+f}|\widehat{\boldsymbol{x}}_{t-p:t}, \boldsymbol{z})$          ▷ generate next reference motion segment
12:             **end if**
13:             $\boldsymbol{s}_t \leftarrow (\boldsymbol{x}_{t-1}, \widehat{\boldsymbol{x}}_{t-1}, \boldsymbol{z}); \ \boldsymbol{a}_t, \widetilde{\boldsymbol{a}}_t \leftarrow \widetilde{\pi}_\theta(\boldsymbol{a}_t, \widetilde{\boldsymbol{a}}_t|\boldsymbol{s}_t)$
14:             $\boldsymbol{x}_t \leftarrow$ next state from simulation with $\boldsymbol{a}_t$ and $\widetilde{\boldsymbol{a}}_t$
15:             $r_t \leftarrow$ reward from Eq. (3) or (5)
16:             $\boldsymbol{s}_{t+1} \leftarrow (\boldsymbol{x}_t, \widehat{\boldsymbol{x}}_t, \boldsymbol{z})$
17:             store $(\boldsymbol{s}_t, \boldsymbol{a}_t, \widetilde{\boldsymbol{a}}_t, r_t, \boldsymbol{s}_{t+1})$ into memory $\mathcal{D}$
18:         **end for**
19:     **end while**
20:     $\theta \leftarrow$ PPO [44] update using trajectory samples in $\mathcal{D}$          ▷ update control policy $\widetilde{\pi}_\theta$
21: **end while**

---

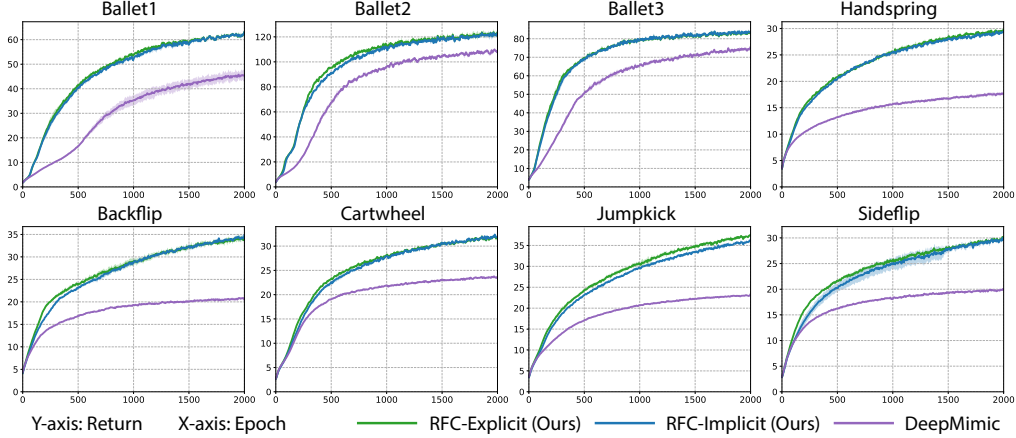

Figure 2: Learning curves of our RFC models and DeepMimic for imitating various agile motions.

The learning procedure for the control policy $\widetilde{\pi}_\theta$ is outlined in Alg. 1. In each RL episode, we autoregressively apply the kinematic policy $n$ times to generate reference motions $\widehat{\boldsymbol{x}}_{p:p+nf}$ of $nf$ steps, and the agent with policy $\widetilde{\pi}_\theta$ is rewarded for imitating $\widehat{\boldsymbol{x}}_{p:p+nf}$. The reason for autoregressively generating $n$ segments of future motions is to let the policy $\widetilde{\pi}_\theta$ learn stable transitions through adjacent motion segments (e.g., $\widehat{\boldsymbol{x}}_{p:p+f}$ and $\widehat{\boldsymbol{x}}_{p+f:p+2f}$). At test time, we use the kinematic policy $\kappa_\psi$ and control policy $\widetilde{\pi}_\theta$ jointly to synthesize infinite-horizon human motions by continuously forecasting futures with $\kappa_\psi$ and physically tracking the forecasted motions with $\widetilde{\pi}_\theta$.

# 6 Experiments

Our experiments consist of two parts: (1) Motion imitation, where we examine whether the proposed RFC can help overcome the dynamics mismatch and enable the humanoid to learn more agile behaviors from reference motions; (2) Extended motion synthesis, where we evaluate the effectiveness of the proposed dual-policy control along with RFC in synthesizing long-term human motions.

## 6.1 Motion Imitation

**Reference Motions.** We use the CMU motion capture (MoCap) database (link) to provide reference motions for imitation. Specifically, we deliberately select eight clips of highly agile motions to increase the difficulty. We use clips of ballet dance with signature moves like pirouette, arabesque and jeté, which have sophisticated foot-ground interaction. We also include clips of acrobatics such as handsprings, backflips, cartwheels, jump kicks and side flips, which involve dynamic body rotations.

**Implementation Details.** We use MuJoCo [49] as the physics engine. We construct the humanoid model from the skeleton of subject 8 in the CMU Mocap database while the reference motions we use are from various subjects. The humanoid model has 38 DoFs and 20 rigid bodies with properly assigned geometries. Following prior work [41], we add the motion phase to the state of the humanoid agent. We also use the stable PD controller [48] to compute joint torques. The simulation runs at 450Hz and the policy operates at 30Hz. We use PPO [44] to train the policy for 2000 epochs, each with 50,000 policy steps. Each policy takes about 1 day to train on a 20-core machine with an NVIDIA RTX 2080 Ti. More implementation details can be found in the supplementary materials.

**Comparisons.** We compare the two variants – RFC-Explicit and RFC-Implicit – of our approach against the state-of-the-art method for motion imitation, DeepMimic [41]. For fair comparison, the only differences between our RFC models and the DeepMimic baseline are the residual forces and the regularizing reward. Fig. 2 shows the learning curves of our models and DeepMimic, where we plot the average return per episode against the training epochs for all eight reference motions. We train three models with different initial seeds for each method and each reference motion. The return is computed using only the motion imitation reward and excludes the regularizing reward. We can see that both variants of RFC converge faster than DeepMimic consistently. Moreover, our RFC models always converge to better motion policies as indicated by the higher final returns. One can also observe that RFC-Explicit and RFC-Implicit perform similarly, suggesting that they are equally

Table 1: Quantitative results for human motions synthesis.

| Method | Phsyics-based | Human3.6M (Mix) | | Human3.6M (Cross) | | EgoMocap | |
|---|---|---|---|---|---|---|---|
| | | MAE ↓ | FAE ↓ | MAE ↓ | FAE ↓ | MAE ↓ | FAE ↓ |
| RFC-Explicit (Ours) | ✓ | **2.498** | **2.893** | 2.379 | **2.802** | 0.557 | 0.710 |
| RFC-Implicit (Ours) | ✓ | **2.498** | 2.905 | **2.377** | **2.802** | **0.556** | **0.701** |
| EgoPose [57] | ✓ | 2.784 | 3.732 | 2.804 | 3.893 | 0.922 | 1.164 |
| ERD [9] | ✗ | 2.770 | 3.223 | 3.066 | 3.578 | 0.682 | 1.092 |
| acLSTM [25] | ✗ | 2.909 | 3.315 | 3.386 | 3.860 | 0.715 | 1.130 |

capable of imitating agile motions. Since the motion quality of learned policies is best seen in videos, we encourage the reader to refer to the supplementary video[1] for qualitative comparisons. One will observe that RFC can successfully imitate the sophisticated ballet dance skills while DeepMimic fails to reproduce them. We believe the failure of DeepMimic is due to the dynamics mismatch between the humanoid model and real humans, which results in the humanoid unable to generate the external forces needed to produce the motions. On the other hand, RFC overcomes the dynamics mismatch by augmenting the original humanoid dynamics with learnable residual forces, which enables a more flexible new dynamics that admits a wider range of agile motions. We note that the comparisons presented here are only for simulation domains (e.g., animation and motion synthesis) since external residual forces are not directly applicable to real robots. However, we do believe that RFC could be extended to a warm-up technique to accelerate the learning of complex policies for real robots, and the residual forces needed to overcome the dynamics mismatch could be used to guide agent design.

## 6.2 Extended Motion Synthesis

**Datasets.** Our experiments are performed with two motion capture datasets: Human3.6M [14] and EgoMocap [57]. Human3.6M is a large-scale dataset with 11 subjects (7 labeled) and 3.6 million total video frames. Each subject performs 15 actions in 30 takes where each take lasts from 1 to 5 minutes. We consider two evaluation protocols: (1) Mix, where we train and test on all 7 labeled subjects but using different takes; (2) Cross, where we train on 5 subjects (S1, S5, S6, S7, S8) and test on 2 subjects (S9 and S11). We train a model for each action for all methods. The other dataset, EgoMocap, is a relatively small dataset including 5 subjects and around 1 hour of motions. We train the models using the default train/test split in the mixed subject setting. Both datasets are resampled to 30Hz to conform to the policy.

**Implementation Details.** The simulation setup is the same as the motion imitation task. We build two humanoids, one with 52 DoFs and 18 rigid bodies for Human3.6M and the other one with 59 DoFs and 20 rigid bodies for EgoMocap. For both datasets, the kinematic policy $\kappa_\psi$ observes motions of $p = 30$ steps (1s) to forecast motions of $f = 60$ steps (2s). When training the control policy $\widetilde{\pi}_\theta$, we generate $n = 5$ segments of future motions with $\kappa_\psi$. Please refer to the supplementary materials for additional implementation details.

**Baselines and Metrics.** We compare our approach against two well-known kinematics-based motion synthesis methods, ERD [9] and acLSTM [25], as well as a physics-based motion synthesis method that does not require task guidance or user input, EgoPose [57]. We use two metrics, mean angle error (MAE) and final angle error (FAE). MAE computes the average Euclidean distance between predicted poses and ground truth poses in angle space, while FAE computes the distance for the final frame. Both metrics are computed with a forecasting horizon of 2s. For stochastic methods, we generate 10 future motion samples to compute the mean of the metrics.

**Results.** In Table 1, we show quantitative results of all models for motion forecasting over the 2s horizon, which evaluates the methods' ability to infer future motions from the past. For all datasets and evaluation protocols, our RFC models with dual-policy control outperform the baselines consistently in both metrics. We hypothesize that the performance gain over the other physics-based method, EgoPose, can be attributed to the use of kine-

Table 2: Ablation Study.

| Component | | Metric | |
|---|---|---|---|
| AddAct | ResForce | MAE ↓ | FAE ↓ |
| ✓ | ✓ | **2.498** | **2.893** |
| ✓ | ✗ | 2.610 | 3.150 |
| ✗ | ✓ | 3.099 | 3.634 |

matic policy and the residual forces. To verify this hypothesis, we conduct an ablation study in the Human3.6M (Mix) setting. We train model variants of RFC-Implicit by removing the residual forces (ResForce) or the additive action (AddAct) that uses the kinematic policy's output. Table 2 demonstrates that, in either case, the performance decreases for both metrics, which supports our previous hypothesis. Unlike prior physics-based methods, our approach also enables synthesizing sitting motions even when the chair is not modeled in the physics environment, because the learned residual forces can provide the contact forces need to support the humanoid. Furthermore, our model allows infinite-horizon stable motion synthesis by autoregressively applying dual policy control. As motions are best seen in videos, please refer to the supplementary video for qualitative results.

## 7 Conclusion

In this work, we proposed residual force control (RFC), a novel and simple method to address the dynamics mismatch between the humanoid model and real humans. RFC uses external residual forces to provide a learnable time-varying correction to the dynamics of the humanoid model, which results in a more flexible new dynamics that admits a wider range of agile motions. Experiments showed that RFC outperforms state-of-the-art motion imitation methods in terms of convergence speed and motion quality. RFC also enabled the humanoid to learn sophisticated skills like ballet dance which have eluded prior work. Furthermore, we proposed a dual-policy control framework to synthesize multi-modal infinite-horizon human motions without any task guidance or user input, which opened up new avenues for automated motion generation and large-scale character animation. We hope our exploration of the two aspects of human motion, dynamics and kinematics, can encourage more work to view the two from a unified perspective. One limitation of the RFC framework is that it can only be applied to simulation domains (e.g., animation, motion synthesis, pose estimation) in its current form, as real robots cannot generate external residual forces. However, we do believe that RFC could be applied as a warm-up technique to accelerate the learning of complex policies for real robots. Further, the residual forces needed to overcome the dynamics mismatch could also be used to inform and optimize agent design. These are all interesting avenues for future work.

## Broader Impact

The proposed techniques, RFC and dual policy control, enable us to create virtual humans that can imitate a variety of agile human motions and autonomously exhibit long-term human behaviors. This is useful in many applications. In the context of digital entertainment, animators could use our approach to automatically animate numerous background characters to perform various motions. In game production, designers could make high-fidelity physics-based characters that interact with the environment robustly. In virtual reality (VR), using techniques like ours to improve motion fidelity of digital content could be important for applications such as rehabilitation, sports training, dance instruction and physical therapy. The learned motion policies could also be used for the preservation of cultural heritage such as traditional dances, ceremonies and martial arts.

Our research on physics-based human motion synthesis combined with advances of human digitalization in computer graphics could be used to generate highly realistic human action videos which are visually and physically indistinguishable from real videos. Similar to the creation of 'deepfakes' using image synthesis technology, the technology developed in this work could enable more advanced forms of fake video generation, which could lead to the propagation of false information. To mitigate this issue, it is important that future research should continue to investigate the detection of synthesized videos of human motions.

## Acknowledgment

This project was sponsored in part by IARPA (D17PC00340).

## Footnotes

[1]Video: https://youtu.be/XuzH1u78o1Y.

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
