[Supplementary Material · rfc_supp.pdf]

# Supplementary Material:
# Residual Force Control for Agile Human Behavior Imitation and Extended Motion Synthesis

## 1 Symbol Tables

Table 1: Important notations used in the main paper.

| Notation | Description |
|---|---|
| $s$ | state |
| $a$ | action |
| $r$ | reward |
| $r^{\text{im}}$ | motion imitation reward |
| $r^{\text{reg}}$ | regularizing reward of residual forces |
| $\gamma$ | discount factor |
| $x$ | humanoid state, $x = (q, \dot{q})$ |
| $\widehat{x}$ | reference humanoid state, $x = (\widehat{q}, \widehat{\dot{q}})$ |
| $x_{0:T}$ | humanoid motion $x_{0:T} = (x_0, \ldots, x_{T-1})$ |
| $\widehat{x}_{0:T}$ | reference motion $\widehat{x}_{0:T} = (\widehat{x}_0, \ldots, \widehat{x}_{T-1})$ |
| $q$ | humanoid DoFs, $q = (q_{\text{r}}, q_{\text{nr}})$ |
| $q_{\text{r}}$ | humanoid root DoFs (global position and orientation) |
| $q_{\text{nr}}$ | humanoid non-root DoFs |
| $\dot{q}$ | joint velocities |
| $\ddot{q}$ | joint accelerations |
| $u$ | target joint angles of PD controllers |
| $\tau$ | joint torques computed from PD control |
| $k_{\text{p}}, k_{\text{d}}$ | PD controller gains |
| $\widetilde{a}$ | corrective action (residual forces) |
| $\mathcal{T}(s_{t+1}\|s_t, a_t)$ | original humanoid dynamics |
| $\widetilde{\mathcal{T}}(s_{t+1}\|s_t, a_t, \widetilde{a}_t)$ | RFC-based humanoid dynamics |
| $\pi_\theta(a_t\|s_t)$ | original humanoid control policy |
| $\widetilde{\pi}_\theta(a_t, \widetilde{a}_t\|s_t)$ | RFC-based composite policy |
| $\widetilde{\pi}_{\theta_1}(a_t\|s_t)$ | humanoid control policy same as $\pi_\theta(a_t\|s_t)$ |
| $\widetilde{\pi}_{\theta_2}(\widetilde{a}_t\|s_t)$ | residual force policy |
| $\xi_j$ | residual force vector |
| $e_j$ | the contact point of $\xi_j$ |
| $h_i$ | contact force vector determined by simulation |
| $v_i$ | the contact point of $h_i$ |
| $J_{e_j}$ | Jacobian matrix $de_j/dq$ |
| $J_{v_i}$ | Jacobian matrix $dv_i/dq$ |
| $B(q)$ | inertial matrix |
| $C(q, \dot{q})$ | matrix of Coriolis and centrifugal terms |
| $g(q)$ | gravity vector |
| $\eta$ | total joint torques $\sum J_{e_j}^T \xi_j$ of residual forces, $\eta = (\eta_{\text{r}}, \eta_{\text{nr}})$ |
| $\eta_{\text{r}}$ | torques of root DoFs from residual forces |
| $\eta_{\text{nr}}$ | torques of non-root DoFs from residual forces |
| $z$ | latent variable for human intent |
| $\kappa_\psi(x_{t:t+f}\|x_{t-p:t}, z)$ | kinematic policy (decoder distribution) in CVAE |
| $q_\phi(z\|x_{t-p:t}, x_{t:t+f})$ | approximate posterior (encoder distribution) in CVAE |

# 2 Motion Imitation Reward

In the following, we give a detailed definition of the motion imitation reward $r_t^{\text{im}}$ introduced in Sec. 3 of the main paper, which is used to encourage the humanoid motion $\boldsymbol{x}_{0:T}$ generated by the policy to match the reference motion $\widehat{\boldsymbol{x}}_{0:T}$. We use two types of imitation reward $r_t^{\text{im}}$ depending on the length of the motion:

(1) World coordinate reward $r_t^{\text{world}}$, which is the reward used in DeepMimic [3]. It has proven to be effective for imitating short clips ($< 5$s) of locomotions, but not suitable for long-term motions due to global position drifts as pointed out in [6]. We still use this reward for imitating short clips of locomotions (e.g., acrobatics) to have a fair comparison with DeepMimic.

(2) Local coordinate reward $r_t^{\text{local}}$, which is the reward used in EgoPose [6]. It is more robust to global position drifts and we use it to imitate longer motion clips (e.g., ballet dance). For extended motion synthesis, we also use $r_t^{\text{local}}$ to imitate the output motion of the kinematic policy $\kappa_\psi$.

Note that for the same reference motion, we always use the same motion imitation reward for both our RFC models and DeepMimic for a fair compairson.

## 2.1 World Coordinate Reward

As defined in DeepMimic [3], the world coordinate reward $r_t^{\text{world}}$ consists of four sub-rewards:

$$r_t^{\text{world}} = w_{\text{p}} r_t^{\text{p}} + w_{\text{v}} r_t^{\text{v}} + w_{\text{e}} r_t^{\text{e}} + w_{\text{c}} r_t^{\text{c}}, \tag{1}$$

where $w_{\text{p}}, w_{\text{v}}, w_{\text{e}}, w_{\text{c}}$ are weighting factors, which we set to (0.3, 0.1, 0.5, 0.1).

The pose reward $r_t^{\text{p}}$ measures the mismatch between joint DoFs $\boldsymbol{q}_t$ and the reference $\widehat{\boldsymbol{q}}_t$ for non-root joints. We use $\boldsymbol{b}_t^j$ and $\widehat{\boldsymbol{b}}_t^j$ to denote the local orientation quaternion of joint $j$ computed from $\boldsymbol{q}_t$ and $\widehat{\boldsymbol{q}}_t$ respectively. We use $\boldsymbol{b}_1 \ominus \boldsymbol{b}_2$ to denote the relative quaternion from $\boldsymbol{b}_2$ to $\boldsymbol{b}_1$, and $\|\boldsymbol{b}\|$ to compute the rotation angle of $\boldsymbol{b}$.

$$r_t^{\text{p}} = \exp\left[-\alpha_{\text{p}}\left(\sum_j \|\boldsymbol{b}_t^j \ominus \widehat{\boldsymbol{b}}_t^j\|^2\right)\right]. \tag{2}$$

The velocity reward $r_t^{\text{v}}$ measures the difference between joint velocities $\dot{\boldsymbol{q}}_t$ and the reference $\widehat{\dot{\boldsymbol{q}}}_t$. The reference velocity $\widehat{\dot{\boldsymbol{q}}}_t$ is computed using finite difference:

$$r_t^{\text{v}} = \exp\left[-\alpha_{\text{v}}\|\dot{\boldsymbol{q}}_t - \widehat{\dot{\boldsymbol{q}}}_t\|^2\right]. \tag{3}$$

The end-effector reward $r_t^{\text{e}}$ measures the difference between end-effector position $\boldsymbol{g}_t^e$ and the reference position $\widehat{\boldsymbol{g}}_t^e$ in the world coordinate:

$$r_t^{\text{e}} = \exp\left[-\alpha_{\text{e}}\left(\sum_e \|\boldsymbol{g}_t^e - \widehat{\boldsymbol{g}}_t^e\|^2\right)\right]. \tag{4}$$

The center-of-mass reward $r_t^{\text{c}}$ encourages the humanoid's center of mass $\boldsymbol{c}_t$ to match the reference $\widehat{\boldsymbol{c}}_t$:

$$r_t^{\text{c}} = \exp\left[-\alpha_{\text{c}}\|\boldsymbol{c}_t - \widehat{\boldsymbol{c}}_t\|^2\right]. \tag{5}$$

The weighting factors $\alpha_{\text{p}}, \alpha_{\text{v}}, \alpha_{\text{e}}, \alpha_{\text{c}}$ are set to (2, 0.005, 5, 100). All the hyperparameters of $r_t^{\text{world}}$ are tuned to achieve best performance for the DeepMimic baseline.

## 2.2 Local Coordinate Reward

The local coordinate reward $r_t^{\text{local}}$ also consists of four sub-rewards:

$$r_t^{\text{local}} = w_{\text{p}} r_t^{\text{p}} + w_{\text{e}} r_t^{\text{e}} + w_{\text{rp}} r_t^{\text{rp}} + w_{\text{rv}} r_t^{\text{rv}}, \tag{6}$$

where $w_{\text{p}}, w_{\text{e}}, w_{\text{rp}}, w_{\text{rv}}$ are weighting factors set to (0.5, 0.3, 0.1, 0.1) same as EgoPose [6].

The pose reward $r_t^{\text{p}}$ is the same as that in $r_t^{\text{world}}$ as defined in Eq. (2). The end-effector reward $r_t^{\text{e}}$ takes the same form as Eq. (4) but the end-effector positions $\boldsymbol{g}_t^e$ and $\widehat{\boldsymbol{g}}_t^e$ are computed in the humanoid's

local heading coordinate. The root pose reward $r_t^{\mathtt{rp}}$ encourages the humanoid's root joint to have the same height $y_t$ and orientation quaternion $\boldsymbol{o}_t$ as the reference $\widehat{y}_t$ and $\widehat{\boldsymbol{o}}_t$:

$$r_t^{\mathtt{rp}} = \exp\left[-\alpha_{\mathtt{rp}}\left((y_t - \widehat{y}_t)^2 + \|\boldsymbol{o}_t \ominus \widehat{\boldsymbol{o}}_t\|^2\right)\right]. \tag{7}$$

The root velocity reward $r_t^{\mathtt{rv}}$ penalizes the deviation of the root's linear velocity $\boldsymbol{l}_t$ and angular velocity $\boldsymbol{\omega}_t$ from the reference $\widehat{\boldsymbol{l}}_t$ and $\widehat{\boldsymbol{\omega}}_t$:

$$r_t^{\mathtt{rv}} = \exp\left[-\|\boldsymbol{l}_t - \widehat{\boldsymbol{l}}_t\|^2 - \alpha_{\mathtt{rv}}\|\boldsymbol{\omega}_t - \widehat{\boldsymbol{\omega}}_t\|^2\right]. \tag{8}$$

Note that all features are computed in the local heading coordinate of the humanoid instead of the world coordinate. The weighting factors $\alpha_{\mathtt{p}}, \alpha_{\mathtt{e}}, \alpha_{\mathtt{rp}}, \alpha_{\mathtt{rv}}$ are set to (2, 20, 300, 0.1) same as EgoPose [6].

# 3 Additional Implementation Details

**Residual Forces.** In RFC-Explicit, each $\boldsymbol{\xi}_j$ is a 6 dimensional vector including both the force and torque applied at the contact point $\boldsymbol{e}_j$. The torque is needed since it along with the force can model the total effect of multiple forces applied at different contact points for a single rigid body of the humanoid. We scale $\boldsymbol{\xi}_j$ by 100 after it is output by the policy. In RFC-Implicit, we similarly scale the total root torques $\boldsymbol{\eta}_{\mathtt{r}}$ by 100. The weight $w_{\mathtt{reg}}$ for the regularizing reward $r_t^{\mathtt{reg}}$ defined in Eq. (3) and Eq. (5) of the main paper is set to 0.1 for both RFC-Explicit and RFC-Implicit. For RFC-Expicit, $k_{\mathtt{f}}$ and $k_{\mathtt{cp}}$ are set to 1 and 4 respectively. For RFC-Implicit, $k_{\mathtt{r}}$ is set to 1.

**Time Efficiency.** Without the residual forces, the total time of a single policy step with simulation is 3.9ms. After adding the residual forces, the time is 4.0ms for RFC-Explicit and 4.3ms for RFC-Implicit. The slight increase in time is due to the larger action space of RFC as well as the Jacobian computation in RFC-Explicit. The processing time 4.0ms translates to 250 FPS, which is well above the interactive frame rate, and the performance gain from RFC justifies the small sacrifice in speed.

**Humanoid Model.** As mentioned in the main paper, we have different humanoid models for different datasets, and the number of DoFs and rigid bodies varies for different humanoid models. Each DoF except for the root is implemented as a hinge joint in MuJoCo. Most joints have 3 DoFs meaning 3 consecutive hinge joints that form a 3D rotation parametrized by Euler angles. Only the elbow and knee joints have just one DoF. In MuJoCo, there are many parameters one can specify for each joint (e.g., stiffness, damping, armature). We only set the armatrue inertia to 0.01 to further stablize the simulation. We leave the stiffness and damping to 0 since they are already modeled in the gains $k_{\mathtt{p}}$ and $k_{\mathtt{d}}$ of the PD controllers. The gains $k_{\mathtt{p}}$ range from 200 to 1000 where stronger joints like spine and legs have larger gains while weaker joints like arms and head have smaller gains. The gains $k_{\mathtt{d}}$ are set to $0.2k_{\mathtt{p}}$. We also set proper torque limits ranging from 50 to 200 based on the gains to prevent instability. In our experiments, we find that the learned motion policies are not sensitive to the gains and scaling the gains by reasonable amount yields similar results. We believe the reason is that the policy can learn to adjust to different scales of gains.

## 3.1 Motion Imitation

The reference motions we use are from the following motion clips in the CMU MoCap database[1]: 05_06 (ballet1), 05_07 (ballet2), 05_13 (ballet3), 88_01 (backflip), 90_02 (cartwheel), 90_05 (jump kick), 90_08 (side flip), 90_11 (handspring). The motion clips are downsampled to 30Hz. We do not apply any filters to smooth the motions and directly use the downsampled motions for training.

Table 2: Training hyperparameters for motion imitation.

| Parameter | $\gamma$ | GAE($\lambda$) | Batch Size | Minibatch Size | Policy Stepsize | Value Stepsize | PPO clip $\epsilon$ |
|---|---|---|---|---|---|---|---|
| Value | 0.95 | 0.95 | 50000 | 2048 | $5 \times 10^{-5}$ | $3 \times 10^{-4}$ | 0.2 |

**Training.** In each RL episode, the state of the humanoid agent is initialized to the state of a random frame in the reference motion. The episode is terminated when then end of the reference motion

Figure 1: Network architectures for the CVAE encoder $q_\phi$ and kinematic policy (decoder) $\kappa_\psi$.

is reached or the humanoid's root height is 0.1 below the minimum root height in the reference motion. As discussed in the main paper, the control policy $\pi_\theta$ is a Gaussian policy whose mean $\boldsymbol{\mu}_\theta$ is generated by a multi-layer perceptron (MLP). The MLP has two hidden layers (512, 256) with ReLU activations. The diagonal elements of the policy's covarian matrix $\boldsymbol{\Sigma}$ are set to 0.1. We use the proximal policy optimization (PPO [5]) to learn the policy $\pi_\theta$. We use the generalized advantage estimator GAE($\lambda$) [4] to compute the advantage for policy gradient. The policy is updated with Adam [2] for 2000 epochs. The hyperparameter settings are available in Table 2.

## 3.2 Extended Motion Synthesis

**Network Architectures.** The CVAE Encoder distribution $q_\phi(\boldsymbol{z}|\boldsymbol{x}_{t-p:t}, \boldsymbol{x}_{t:t+f}) = \mathcal{N}(\boldsymbol{\mu}_\mathsf{e}, \mathrm{Diag}(\boldsymbol{\sigma}_\mathsf{e}^2))$ is a Gaussian distribution whose parameters $\boldsymbol{\mu}_\mathsf{e}$ and $\boldsymbol{\sigma}_\mathsf{e}$ are generated by a recurrent neural network (RNN) as shown in Fig. 1 (Left). Similarly, the kinematic policy (decoder) $\kappa_\psi(\boldsymbol{x}_{t:t+f}|\boldsymbol{x}_{t-p:t}, \boldsymbol{z}) = \mathcal{N}(\tilde{\boldsymbol{x}}_{t:t+f}, \beta\boldsymbol{I})$ is also a Gaussian distribution whose mean $\tilde{\boldsymbol{x}}_{t:t+f}$ is generated by another RNN as illustrated in Fig. 1 (Right), and $\beta$ is a hyperparameter which we set to 10. We use GRUs [1] as the recurrent units for both $q_\phi$ and $\kappa_\psi$. For the RFC-based control policy $\tilde{\pi}_\theta(\boldsymbol{a}, \tilde{\boldsymbol{a}}|\boldsymbol{x}, \hat{\boldsymbol{x}}, \boldsymbol{z})$, we concatenate $(\boldsymbol{x}, \hat{\boldsymbol{x}}, \boldsymbol{z})$ together and input them to an MLP to produce the mean of the actions $(\boldsymbol{a}, \tilde{\boldsymbol{a}})$. The MLP has two hidden layers (512, 256) with ReLU activations.

Table 3: CVAE Training hyperparameters for the kinematic policy $\kappa_\psi$.

| Parameter | Dim($\boldsymbol{z}$) | Batch Size | Minibatch Size | Initial Learning Rate | KL Tolerance |
|---|---|---|---|---|---|
| Value | 128 | 10000 | 256 | $1 \times 10^{-3}$ | 10 |

**Training.** The kinematic policy $\kappa_\psi$ is trained with the CVAE objective in Eq. (6) of the main paper. We jointly optimize $q_\phi$ and $\kappa_\psi$ for 200 epochs using Adam [2] with a fixed learning rate. We then continue to optimize the model for another 800 epochs while linearly decreasing the learning rate to 0. We list the hyperparameters for training the CVAE in Table 3. The training procedure for the RFC-based control policy $\tilde{\pi}_\theta(\boldsymbol{a}, \tilde{\boldsymbol{a}}|\boldsymbol{x}, \hat{\boldsymbol{x}}, \boldsymbol{z})$ is outlined in Alg. 1 of the main paper. Similar to motion imitation, we optimize the policy $\tilde{\pi}_\theta$ with PPO for 3000 epochs using Adam. The hyperparameter settings are the same as motion imitation as shown in Table 2.

## Footnotes

[1] http://mocap.cs.cmu.edu/