[Reviews · NeurIPS 2020]

Review 1

Summary and Contributions: The paper presents an approach to the imitation learning of physics-based motions for humanoid characters. In contrast to existing methods, the approach presented here does not only learn a policy for generating the control forces for the humanoid. Rather, the approach also used a so-called residual force policy which acts as invisible strings tugging on the humanoid. More specifically the RF-policy generates additional forces which are applied on different locations on the body. As a result, the humanoid can mimic demonstrated movements even in the presence of large (physical) variations between the human demonstrator and the imitating humanoid. The paper also describes a method for motion synthesis in which a conditional variational autoencoder is used to generate future kinematic targets poses. In turn, the RFC-based control policy can then execute actions to reach these target positions. The objective of the motion synthesis process is to generate long-term motion clips in an automatic fashion without the necessity for human inputs. The paper argues that highly complex human motions such as ballet moves (e.g. pirouette) can be imitated using this methodology. Comparisons to SotA methods for physics-based and kinematics-based motion generation techniques are also provided.

Strengths: The approach is novel and quite creative in solving the challenges surrounding the imitation of human mo-cap data within a physics-based simulation. The idea of learning a policy for generating auxilliary, residual forces is really clever and beautifully executed. In my opinion, that seems to be the core contribution of the paper. The approach is rather rigorous in its integration of physics, kinematics, and machine learning within one appealing framework that does not appear "hacked". Given the description I also think that it is straight-forward to implement. In general, the authors have been quite transparent by adding substantial detail in the supplementary material. The results shown in the videos are definitely impressive. Some of the claims made in the paper, however, do not seem sound or are at least odd (more below). The experimental section features a number of comparisons to current SotA method which is commendable. But there are also some questions regarding the experiments and their ability to provide a full picture of the abilities of the proposed approach. One interesting future avenue for this research is to use it as a form of regularization for learning complex policies. In the beginning, high residual forces are allowed to keep the agent stable. With improving performance, these forces can then be annealed down to zero. That could be a very interesting experiment to perform!

Weaknesses: The biggest weakness of the paper is that it seems to be making claims that are not backed up by the results. For example, the paper states that "show for the first time humanoid control policies thatare capable of highly agile ballet dancing moves" and basically argues that policies of similar complexity have not been learned before. However, the DeepMimic paper showed really complex motor skills including back-flips, a pirouette-like spin, Karate moves and other motions that are at least on a similar level of complexity. Further, the paper mostly focuses on tracking the observed motions (potentially small variations thereof). But in imitation learning it is very important to show the generalization capabilities of a learned policy; otherwise a trajectory would suffice. Any experiments in this direction are missing in the paper. To take DeepMimic as an example again: in that paper they demonstrate how a learned kick generalizes to different goal positions. Similarly, they show how a learned walking behavior generalizes to different terrains. It would help if the authors limit the scope of their claims or describe whether and how generalization is achieved. A critical limitation of this approach compared to others is that it is only applicable to simulation environments. Imitation learning is widely used in robotics, cyber-physical systems and other hardware implementations. The specific imitation strategy described here could not be implemented in those domains. That substantially limits the scope and relevance of this technique. That being said, as mentioned above, it would be interesting to see this applied as a form of scaffolding technique for training robots in simulation. One aspect that is not well-explained in the paper is multiplicity of solutions for the residual forces. Meaning, there are many different ways of how to apply stabilizing forces to a human body. Which one of these solutions you will get is not clear. Especially, when synthesizing variations of the imitated motion you may end up with really strange behavior if the residual forces are applied the wrong way, i.e., objects could fly, human could be torn apart, etc. Finally, in my opinion the motion synthesis is not really a novel contribution. Effectively a cVAE is used to predict trajectories which are then followed. This is pretty straight forward stuff and many modern papers use cVAE for exactly that purpose, i.e., trajectory prediction. The paper should really just focus on the RFC part since that is definitely novel and creative.

Correctness: The methods is correct, but some of the claims made are too general and need to be clarified, i.e., the scope limited. In particular the comparison to DeepMimic should be refined. DeepMimic is capable of generalization that is not shown here. In addition, deep mimic is working within the confines of what the humanoid agent can physically do. Hence a comparison of "who learns faster" between RFC and DeepMimic is really unfair since RFC agents are effectively pulled from strings. That being said, I understand that such comparisons are difficult to construct.

Clarity: The paper is very well written and extremely easy to understand. The authors did an excellent job!

Relation to Prior Work: As described above some of the claims regarding the superior performance of the new method are not correct in my opinion. These claims are actually not necessary to make a case for your method. In the motion synthesis part, it would be important to add citations to recent papers on trajectory prediction with cVAE. There are a vast number of papers on that topic.

Reproducibility: Yes

Additional Feedback: ---- Update after author responses ----- I would like to thank the authors for the very helpful responses. Overall, I am very happy with the answers and they helped me better understand potential applications even outside of simulation. I would recommend that the authors used the additional space in the final version to provide more information on how this could be used as a scaffolding technique in robot learning. After carefully thinking about it, I have updated my score slightly based on the author responses.


Review 2

Summary and Contributions: This paper proposes a reinforcement learning based method for humanoid agent control in a physical environment that learns to imitate complex human behaviors. The premise of the proposed method is that traditional humanoid agent control methods do not take into account the difference in transition dynamics between humans and humanoid agents. Therefore, they propose to compensate these dynamic differences by a residual force control (RFC) which augments a humanoid control policy with additional forces to match the human dynamics. In addition, they propose the use of an additional kinematic model that generates infinite horizon human motion which is used to drive the humanoid agent policy. In experiments, they quantitatively and qualitatively outperform previous humanoid control methods.

Strengths: + A new framework for humanoid agent control via imitation. + Outperforms previous methods. + Well written paper

Weaknesses: - On including external forces that are not part of the original humanoid design: I would first like to acknowledge that the idea of having additional forces help the humanoid agent match the dynamics of a more complex human body is very nice and the results look amazing. However, I am having difficulty extrapolating this concept to a more general setup. The first example that comes in mind is an agent that **should** have a difficult time following certain motions. For example, an agent A which may be able to generate less torque than agent B should not be able to do certain moves / tasks, but in this setup, the additional forces will help agent A do what only agent B is able to do. This will take away from the realism component of the motions which agent A can perform. The way I understand this method is to be a type of training wheel agents are able to have to achieve the more complex human motion, however, these training wheels should be removed eventually. Another example is of extending this method to help agents interact with the world around them where the residual forces might intervene while the agent is, say, pushing a block to be able to push it as the imitation source (human or another more complex agent). I would appreciate it if the authors can comment on this, and please point out if I am missing something. - Kinematic policy naming: I feel the naming of “kinematic policy” is not completely accurate. As far as I know, we refer to policies as functions that map states to actions. In this case, the kinematic policy looks more like a conditional generative model of human motion (i.e., Equation 5). In addition, the overall method proposed in this paper is more like a model based reinforcement learning approach for imitation where the model is the human motion synthesis model previously mentioned. I would recommend the authors to rename the kinematic policy to something like “kinematic model” to prevent any confusions where readers may expect an actual policy. - Additional evaluations: Since there is a generative model of human motion, it would be good to present an evaluation of the diversity of generation for the models trained with CMU Mocap and Human 3.6M. I can see multiple possible futures in the supplementary video, but numbers would be good to have too.

Correctness: Not sure about using residual forces not present in the original humanoid agent, but I am looking forward to the authors response to this concern. Other than that, the method seems correct to me.

Clarity: Yes

Relation to Prior Work: Yes, but some relevant previous work is missing: Character control: http://homepages.inf.ed.ac.uk/tkomura/dog.pdf http://www.ipab.inf.ed.ac.uk/cgvu/nsm.pdf Human motion synthesis: http://proceedings.mlr.press/v70/villegas17a/villegas17a.pdf https://openaccess.thecvf.com/content_ECCV_2018/papers/Xinchen_Yan_Generating_Multimodal_Human_ECCV_2018_paper.pdf

Reproducibility: Yes

Additional Feedback: In conclusion, I think this is a really nice paper with excellent results. My current assessment is of slightly leaning towards acceptance because of my initial concern about the residual forces in a more general setting. I am open to increasing my score once the authors have addressed this, and the additional feedback I provided. ###### POST REBUTTAL FEEDBACK ###### After reading the rebuttal and other reviewers comments, I decided to keep my score. I really like this paper as it is for applications in physical simulations such as physics based video games. However, I am concerned about the "invisible hands constantly interacting with the humanoid", as R3 put it, in real world robotics. Having said that, it looks like the authors are working on that exact problem so this work looks a promising intuitive way for enabling RL humanoid models to more easily learn movement policies regardless of different body dynamics.


Review 3

Summary and Contributions: - The paper proposes new methods for motion imitation and extended motion synthesis based on two ideas (1) residual force control which allows the agent to add imaginary fake forces at contact points (explicit) or directly at (actuated or unactuated) joints (implicit), and (2) conditional VAE-based kinematic policy that can predict future segments given the past segments, allowing tracking-based policy to be used for recursive, long-horizon motion synthesis. - Empirically, the paper demonstrates substantially better imitation policies for difficult ballet movements compared to DeepMimic in imitation setting, and outperforms other methods in extended motion synthesis. Interestingly, the method works for extended motion synthesis for scenes that are physically impossible (sitting motions in space without chairs).

Strengths: - Strong empirical results: The numerical evaluations and videos are quite extensive in demonstrating the benefits of this approach over the prior methods. The results can certainly advance the state of data-driven complex motion synthesis. However, I’d classify this as a strong application paper. Residual forces have been well explored in prior motion synthesis literature (e.g. CIO) in computer graphics. - Clear writing: The paper is written very cleanly (motivation, methods, empirical validations). The sections on control are also suitable for the general ML audience. It’s a high-quality paper. - Sensible algorithmic design choices: RFC explicit vs implicit, the choice of action space, the use of CVAE were all explored well. The paper contains substantial insights for future work on better motion synthesis.

Weaknesses: - Physics violation: line 155 “T needs to be physically-valid and respect physical constraints (e.g., contacts)”. With external residual forces, the physics is invalid (unless in the real world you can have some invisible hands constantly interacting with the humanoid). Adding “some” could qualify this sentence (e.g. respect “some” physical constraints), because it is violating some physics. Additionally, it will be helpful to include comparisons with the option of directly controlling state deltas using some prior knowledge on deltas per step (can be thought of as teleportation). While I agree that with RFC it’s easier to quantify how much physics is violated (Eq 2 and 4) and arguably more interpretable, including this baseline could be nice, since such an approach will not require hacking physics simulator like Mujoco and can make such physics-violating motion synthesis method applicable to more domains. The comparisons could be to evaluate if such state delta-based indeed produce less realistic movements than RFC methods, which I am uncertain is true without actual experiments. - Generalizations and robustness of the learned policy: if during training the initial states are set deterministically to states, this could hurt generalization (just memorizing a single trajectory and not knowing how to act outside). Have authors experimented with randomization in training and testing? Extended motion synthesis may prove some robustness but it will be good to include such metrics directly in Motion Imitation experiments.

Correctness: - The methods and evaluation protocols are correct. The claims (except some listed on weaknesses) are well substantiated.

Clarity: - The paper is written very cleanly (motivation, methods, empirical validations). The sections on control are also suitable for the general ML audience. It’s a high-quality paper.

Relation to Prior Work: - The paper discusses and compares to several prior works in motion imitation and extended motion synthesis. It could benefit from discussing more motion synthesis methods that violate physics during optimization. For example, direct collocation methods (e.g. CIO [1] [2]) should be cited. Those approaches solve trajectory optimization through constrained formulation, which can relax physics during optimization. These methods, however, aim to eventually produce solutions that satisfy physics (constraints). It’s also good to consider if this can also be applied in your setting (e.g. using external forces to aim learning, but eventually driving them to 0). Esp RFC-Implicit Eq 3 is closely related to regularization in CIO etc. [1] Mordatch, Igor, Emanuel Todorov, and Zoran Popović. "Discovery of complex behaviors through contact-invariant optimization." ACM Transactions on Graphics (TOG) 31.4 (2012): 1-8. [2] Mordatch, Igor, Zoran Popović, and Emanuel Todorov. "Contact-invariant optimization for hand manipulation." Proceedings of the ACM SIGGRAPH/Eurographics symposium on computer animation. 2012.

Reproducibility: Yes

Additional Feedback: Post rebuttal: Thanks to the authors for answering questions and adding additional discussions. It's a great submission and other reviewers also agree. --- - Not very related but this paper also applies adaptive external forces (for automatic curriculum learning) [3]. - line 330: “Unlike prior physics-based methods, our approach also enables synthesizing sitting motions even when the chair is not modeled in the physics environment, because the learned residual forces can provide the contact forces needed to support the humanoid.” I quite like this observation and enjoyed seeing it in the video. Having more discussion in the main text could be valuable. [3] Pinto, Lerrel, et al. "Robust adversarial reinforcement learning." arXiv preprint arXiv:1703.02702 (2017).


Review 4

Summary and Contributions: In this paper, the authors introduced an extra learned contact force policy, or equivalently, a learned extra force/torque policy to the center of mass of the simulated character. With this extra policy, they are able to train complex, long-ranged, agile human motion, such as ballet dancing using motion imitation data. The performance and sample efficiency of their approach beat the state of the art methods such as Deep Mimic.

Strengths: The results from this paper looks amazingly good. Also, the authors introduced a third kinematic policy, which allows them to arbitrarily compose sequence of complex motions and realize these on a physical embodiment.

Weaknesses: Although the results look amazing, the baseline chosen (Deep Mimic) is actually not the newest and more powerful MCP (Multiplicative Compositional Policies) method. I noticed that the authors have cited the MCP paper, and I encourage them to actually use that as an additional baseline for comparison. Also, so far all benchmarks have been limited to human motions. I think adding different types of non-human characters in experiments would be very useful.

Correctness: The authors introduced regularization to the learned contact/CoM forces during training. However, there is no hard constraints on whether the learned forces are physically admissible. For example, with a soft regularization, the contact forces can be negative (i.e. pulling contact bodies together), or non-zero when the two bodies are separating. In the implicit formulation, the physics can be completely violated as the extra CoM forces can inject lots of energy. More discussions and validations on this aspects are needed.

Clarity: The paper is clearly written.

Relation to Prior Work: Yes.

Reproducibility: Yes

Additional Feedback:

[Author Response · NeurIPS 2020]

We'd like to first thank the reviewers for their constructive feedback. We are grateful for the appreciation of the
novelty of the proposed residual force control (RFC) framework and its empirical effectiveness. We also appreciate the
suggested references and will include them. Here we aim to address the main questions raised by the reviewers.

**(R1) Q1: Motion complexity claims not backed up by results.** A: We will tone down the claims w.r.t. DeepMimic
in the final version, but we'd like to stress that the improvement over DeepMimic for ballet dance motions are backed by
both quantitative (Fig. 2) and qualitative results (video). This is not to take anything away from DeepMimic, but to show
that by compensating for the dynamics mismatch with residual forces, one can achieve better motion imitation. We will
follow R1's suggestions to put comparison with DeepMimic in context and discuss the differences more carefully.

**(R1) Q2: Generalization of RFC and novelty of motion synthesis.** A: The human motion synthesis experiments
are designed specifically to show the generalization of RFC. Note that during test time the RFC policy will **not** be
further trained to imitate the kinematic output, and it is basically tasked to do one-shot imitation. At test time, the
motions generated by the kinematic policy are different from training (different data) and as input to the state of the
RFC policy they are analogous to the goals in DeepMimic. Without the residual forces, the agent often falls to the
ground as demonstrated in the video (please see posing and walking dog motions starting from 3:31). This in fact
demonstrates the generalization of RFC since it is more robust to motion variation than w/o RFC. The key contribution
of the motion synthesis is not about the cVAE but the message that an RFC policy can one-shot imitate the noisy output
of a generative motion prediction model and is able to do so on a large motion dataset (Human3.6M).

**(R1) Q3: Can only be applied to simulation.** A: Although the focus of this paper is on virtual human motion synthesis,
as mentioned by R1 and other reviewers (R2 & R3), the method could be extended to a scaffolding technique for training
complex motion policies, which is actually a direction we are already investigating. Another interesting direction is to
use the residual forces to quantify the dynamics mismatch, which could be used to inform agent design or even uncover
hidden physical objects in the scene (please see the sitting motion in the video starting from 3:09).

**(R1) Q4: Multiplicity of solutions.** A: For RFC-Explicit it is indeed possible to have multiple solutions since there are
excessive DoFs when using multiple residual forces. However, RFC-Implicit introduces minimal new action dimensions
and the regularization reward further reduces the set of feasible solutions. The question on multiplicity of solutions also
applies to imitation learning in general since its only difference with RFC-Implicit is the root actuation.

**(R2) Q5: Residual forces not in the original agent design.** A: For human motion imitation, we are exactly trying to
let an agent do something it is not able to do using the residual forces (RFs), since the agent is much simpler than real
humans. If we don't want the agent to go beyond its ability, then RFC could be extended to a scaffolding technique
(please see Q3). Also, as shown in the video, when the agent is forced to imitate demonstrations from other agents (e.g.,
human), the learned motions without RFs won't be more real. Instead, the agent often fails to imitate the motion or just
falls to the ground. Finally, for agent-object interaction, the RFs won't hinder learning since the policy can always learn
to output 0 RFs if they don't help. The RFs are only applied to stabilize the agent without changing object contact.

**(R2) Q6: Additional evaluation.** A: Since the motion synthesis baselines are deterministic, i.e., no diversity (we
choose them because they allow stable long-term prediction), it would be unfair to compare diversity with them. The
diversity of RFC is similar to our cVAE kinematic model (e.g., 5.7 (cVAE) vs. 5.6 (RFC) for average pairwise sample
distance). Besides, the design of the cVAE itself is not the focus of the paper and can be replaced by other models.

**(R3) Q7: Relation to CIO.** A: These works are indeed related and we will include a discussion. However, we'd like to
stress three key differences: (1) The residual forces (RFs) are **not** the contact forces (CFs) in CIO. RFs are **residuals**
to existing CFs in the simulation which makes them easier to learn and directly quantify the physics violation. (2)
Learning a policy that outputs RFs with RL is novel which can generalize to test data as shown in the motion synthesis
experiments, i.e., no need to solve a trajectory optimization again. (3) Our RFC-Implicit formulation differs from the
CIO physics regularization in that it isolates the physics violation to the root actuation (physically-valid w/o it). This
again shows RFs differ from CFs since RFs are exact changes to physics decoupled from existing CFs in the simulation.

**(R3) Q8: Comparisons with directly controlling state deltas.** A: We did try this approach but it has a major problem:
the delta changes in states do not go through physics simulation and often break contact, leading to unstable simulation.

**(R3) Q9: Robustness of the learned policy.** A: During training the RL policy injects Gaussian noise into the states,
which allows the policy to explore nearby states around the demonstration as well, so the policy is robust to noise.

**(R4) Q10: Comparison with MCP.** A: MCP is a hierarchical method to combine different primitive skills to achieve
tasks. As mentioned in the MCP paper, the primitive skills are still learned through DeepMimic. We used DeepMimic
as baseline since it provides a pure comparison for motion imitation without the extra level of complexity.

**(R4) Q11: No hard constraints on learned forces.** A: The learned residual forces (RFs) are not to replace the contact
forces (please see Eq. 1) but to correct the contact forces and compensate for the dynamics mismatch. So it is still valid
for the RFs to be negative since the sum of the RFs and contact forces won't be negative. Further, the CoM forces are
injecting energy only when they are needed to compensate for the dynamics mismatch (e.g., incorrect mass distribution)
and match the demonstration. Finally, we can also impose hard constraints on the RFs by thresholding their magnitude.

[Meta-Review · NeurIPS 2020]

The reviewers highly appreciated the rebuttal which successfully addressed many doubts. While the general idea of adding external forces to aid the learning process has been previously in other settings, the application to imitation learning for humanoid agents is certainly innovative, the results are very impressive, and the rebuttal convincingly argues for the potential of application beyond simulations.